# The bilevel chamber revealed differential involvement of vasopressin and oxytocin receptors in female mouse sexual behavior

Himeka Hayashi[1¤a], Kie Shimizu[2,3¤b], Kazuaki Nakamura[2,3], Katsuhiko Nishimori[4], Yasuhiko Kondo[1]*

1 Department of Animal Sciences, Teikyo University of Science, Yamanashi, Japan, 2 Division of Life Science, Graduate School of Science and Engineering, Saitama University, Saitama, Japan, 3 Department of Pharmacology, National Research Institute for Child Health and Development, Tokyo, Japan, 4 Department of Obesity and Internal Inflammation, Fukushima Medical University, Fukushima, Japan

¤a Current address: Graduate School of Environmental, Life, Natural Science and Technology, Okayama University, Okayama, Japan
¤b Current address: Department of Animal Science and Biotechnology, Azabu University, Kanagawa, Japan
* ykondo@ntu.ac.jp

**Data Availability Statement:** All relevant data for this study are publicly available from the figshare repository (https://doi.org/10.6084/m9.figshare.25880005.v1).

## Abstract

Arginine vasopressin (AVP) and oxytocin (OT) are well-known as neuropeptides that regulate various social behaviors in mammals. However, little is known about their role in mouse female sexual behavior. Thus, we investigated the role of AVP (v1a and v1b) and OT receptors on female sexual behavior. First, we devised a new apparatus, the bilevel chamber, to accurately observe female mouse sexual behavior. This apparatus allowed for a more precisely measurement of lordosis as receptivity and rejection-like behavior (newly defined in this study), a reversed expression of proceptivity. To address our research question, we evaluated female sexual behavior in mice lacking v1a (aKO), v1b (bKO), both v1a and v1b (dKO), and OT (OTRKO) receptors. aKO females showed decreased rejection-like behavior but a normal level of lordosis, whereas bKO females showed almost no lordosis and no change in rejection-like behavior. In addition, dKO females showed normal lordosis levels, suggesting that the v1b receptor promotes lordosis, but not necessarily, while the v1a receptor latently suppresses it. In contrast, although OTRKO did not influence lordosis, it significantly increased rejection-like behavior. In summary, the present results demonstrated that the v1a receptor inhibits proceptivity and receptivity, whereas the v1b and OT receptors facilitate receptivity and proceptivity, respectively.

## Introduction

Arginine vasopressin (AVP) and oxytocin (OT) are highly evolutionarily conserved humoral hormones secreted from the posterior lobe of the pituitary which also act as neuromodulators regulating various sociosexual behaviors via direct secretion in the central nervous system of fish [1], amphibians [2], birds [3], and mammals [4]. Because peripheral OT regulates parturition and lactation in mammals, its central action was first investigated in maternal behavior,

**Funding:** This research is supported by Grants-in-Aid for Scientific Research from the Japan Society for the Promotion of Science (Y.K.: 21K06274). Please correct this information accordingly.

**Competing interests:** The authors have no conflict of interest to declear.

demonstrating that intracerebroventricular OT administration induced parental nurturing in steroid-primed virgin female rats [5]. This discovery of the central action of OT soon extended to other reproductive behaviors. Either intraperitoneal or intracerebroventricular OT injections increased lordosis, a dorsiflexed response to male mounting, in hormone-primed ovariectomized female rats [6–8]. Local OT application in the medial preoptic area (MPOA) [9, 10], being a part of the inhibitory system for lordosis [11], had a larger lordosis facilitatory effect than in the ventromedial hypothalamus (VMH) [12, 13]. Subject to male mounts, OT immunoreactivity in the MPOA increased in females, while the number of OT immunoreactive cells decreased, suggesting extracellular release of OT in the female MPOA during sexual behavior [10].

On the other hand, research on the neural mechanisms underlying female sexual behavior in mice is scarce. This may be due to the characteristics of mouse sexual behavior, which requires much longer time for both initiation and consummation compared to rats, lacks a typical diagnostic behavior for proceptivity (estrous female rats show characteristic soliciting behavior, such as hopping and darting, and ear-wiggling) and obscure lordosis as receptivity (difficult to distinguish lordosis because of less distinctive characteristics). The concepts of "proceptivity" and "receptivity" were introduced by Beach [14] to delineate components of mammalian female sexual behavior. Receptivity refers to the female's behavior in accepting the male's attempt of copulation, while proceptivity includes various behaviors induce by male stimuli preceding the male's mount. These behavioral components in female mice are ambiguous and have not been clearly defined so far. Nevertheless, the easy gene manipulation in mice provides an advantage for detailed genetic analyses of the neuroendocrine mechanisms underlying sexual behavior. For example, previous studies in female mice deficient in OT or its receptor (OTR) gene demonstrated that OT was indispensable for milk-feeding but not essential for sexual behavior and parturition [15, 16]. Also, it was reported that OT deficiency delays the initiation of sexual behavior in sexually naïve females [17], increases the frequency/duration of nonreceptive posture [18], and decreases the frequency/duration of lordosis *per se* [19]. Moreover, OT and OTR gene deficiencies reportedly have different effects on social behavior [16], so further research is needed to clarify this point.

Since AVP shares an extremely close structure with OT, sharing seven out of nine amino acids, and plays similar functions in social behavior, such as the regulation of social recognition [20, 21], its regulatory role in female sexual behavior has been previously investigated [7, 22–26], and considered more complicated than that of OT. For example, although the intracerebroventricular AVP administration was reported to increase lordosis in female rats [7], another study reported no effect of a similar administration at the same dose and a rather suppressive effect on lordosis at a lower dose [22]. Interestingly, the suppressive effect of central AVP administration on lordosis was facilitatory when combined with OT treatment [23]. Furthermore, the effect of an AVP antagonist on lordosis correlated with AVP concentrations in the suprachiasmatic nucleus, being suppressive at high concentrations and promoting at low [24]. Moreover, AVP infusion in the MPOA increases flank marking behavior in female hamsters but not lordosis [25, 26].

So far, the role of AVP on sexual behavior in female mice has not been investigated. Deletion of AVP-expressing neurons in the paraventricular nucleus of the hypothalamus appears to increase females' social investigatory behavior [27]. AVP and OT directly excite MPOA and VMH neurons [28], and the vasopressin v1a receptor in the lateral septum (LS), including the lordosis inhibitory system [29], was more densely expressed in females than in males [30]. These facts suggest their involvement in female sexual behavior, even in mice.

Therefore, in this study, to facilitate measuring lordosis in female mouse sexual behavior tests, we created a new bilevel apparatus for sexual behavior tests that enables observation

from a side view (Experiment I). Thanks to it, we compared sexual behavior in female mice deficient in vasopressin v1a receptor (aKO), v1b receptor (bKO), both (dKO), and oxytocin receptor (OTRKO) (Experiments II and III). Then, we demonstrated the differential involvement of these receptors in the neural regulation of lordosis and rejection-like behavior during female mouse mating.

## Experiment I: Measurement of female sexual behavior in bilevel chamber

### Materials and methods

**Subjects.** Male and female ICR mice (n = 8) were purchased from Japan SLC Inc. (Shizuoka, Japan). Males and females (3–6 per cage) were separately housed under controlled temperature (23°C ± 1°C) and illumination (08:00–20:00, light on) throughout the experiment. Food and water were provided *ad libitum*. All experiments and animal housing described in this article adhered to the guidelines for the Care and Use of Laboratory Animals of Teikyo University of Science and were approved by the Committee for Experimental Animal Ethics of Teikyo University of Science.

One week before the experiment, all females were ovariectomized under isoflurane anesthesia. In each behavioral test, estrus was induced by subcutaneously injections with 25 μg 17β-estradiol 3-benzoate (EB, Sigma-Aldrich, in 0.05 mL sesame oil) two days before and with 250 μg progesterone (P, Sigma-Aldrich, in 0.05 mL sesame oil) 3–6 h before.

**Mating experience.** Due to the low sexual activity of sexually naïve female mice, we first provided them with mating experience to induce some level of sexual activity in the subsequent bilevel chamber tests. Each female was placed alone in an ordinary observation cage (182 × 260 × 128 mm) and allowed to get adapted to it. Five minutes later, the mating session was started with the introduction of a stimulus male. The observation of sexual behavior was terminated with ejaculation reception or after 30 min, whichever is the first. Female lordosis, and mount, intromission (mount with rhythmic pelvic thrusts) and ejaculation by males were recorded. If a female's lordosis could not be reliably observed, male's intromission was considered to indirectly indicate female lordosis. A lordosis quotient [LQ = (number of lordosis / total number of mounts and intromissions received) × 100] was calculated based on these estimated lordosis occurrences. The mating sessions were repeated weekly until a LQ≥50% was observed. (Indeed, all females met this criterion within three sessions).

**Bilevel chamber test.** In this study, we newly devised a bilevel chamber to observe female mouse sexual behavior. An acrylic box (50 × 24 × 6 cm) was divided at the half height (24 cm) by an opaque board (floor) having square holes (4 × 4 cm) at both ends which connected to the floor below by ladders (Fig 1A). In the apparatus, mice could freely move to the other level, allowing females to easily escape from stimulus males (pacing behavior). The depth of the apparatus was wide enough to move freely for mice, but it constrained them laterally when a male mounted a subject female. This constraint facilitated the experimenter in easily and accurately evaluating whether the female exhibited lordosis. Furthermore, the constraint also made it easier to define rejection-like behavior. In the ordinary observation cage, mice interact with each other by approaching from various directions, leading to diverse behavioral patterns in the rejection of male approaches. However, in this apparatus, the rejection of male approaches could be defined as a typical behavioral pattern because of the limited direction of the approach.

Both the subject females and stimulus males were subjected to the 10–20 min acclimation to the apparatus three times on separate days, to confirm smooth movement between levels. In each test, a subject female was first placed alone in the bilevel chamber, and smooth movement

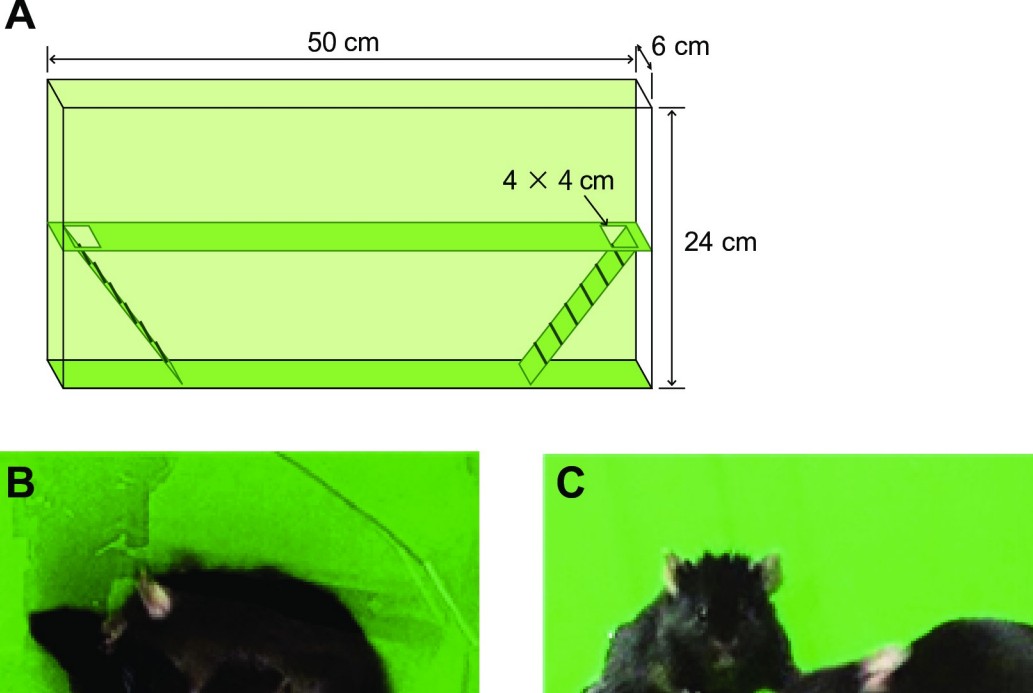

**Fig 1. Bilevel chamber: A new test device for measuring female mouse sexual behavior. (A)** Sketch showing the bilevel chamber apparatus. Ladders connect the first and second levels on both sides. The chamber has a narrow depth of 6 cm to constrain sexual behavior sideways. **(B)** Mouse lordosis was defined as s posture wherein the tail base was lifted to the waist level to present the vagina to the male. We did not judge a situation as lordosis if the female's waist could not be observed because of insufficient sideways turn. (C) Rejection-like behavior was defined as suppression of the male's approach by pressing the male's face or body with her limbs.

was confirmed again during the 5 min acclimation period. Subsequently, when the female was on the lower level, a stimulus male was introduced in the upper level. The test lasted for 20 min or was terminated by male ejaculation, whichever is the first. If no male mount took place within 5 min, the test was restarted with another stimulus male. After completing each behavioral test, the cover in front of the apparatus was opened, and mouse excreta were removed. The inside of the apparatus was sprayed with 70% alcohol and cleaned by wiping with a paper towel.

**Analysis of sexual behavior in the bilevel chamber.** In the bilevel chamber test, we recorded female and male behaviors. Based on their behavioral characteristics, male behaviors were mount, intromission and ejaculation. Female behavior was classified as follows:

**Lordosis:** posture whereby the female raises its tail base to the lumber level to present their vagina to males (Fig 1B). Cases where the tail base was raised from the floor but did not reach the lumber height were excluded from measurements. In addition, cases of insufficient side view to observe lordosis were excluded.

**Rejection-like behavior:** Rejection-like behavior was defined as pressing the male's face and/or body with its limbs to block approaches and mounts (Fig 1C).

Consequently, we classified the bouts of behavioral interactions observed in bilevel chamber tests into five categories:

- •Mount with lordosis

- •Mount without lordosis

- •Intromission with lordosis

- •Intromission without lordosis

- •Rejection-like behavior.

When the behavioral interactions paused > 2 s, it was recorded as another bout. However, if various behavioral patterns such as mount, intromission, or ejaculation continuously occurred within a bout, we determined the category based on the last behavior observed. Then, we recorded the latency, number of occurrences, and duration of each category. After each test, we calculated the intromission ratio [IR = (number of intromissions / total number of mounts and intromissions)] of the stimulus males and the LQ of the subject female.

All behaviors were recorded by a remote video system during behavioral tests. The occurrences and time spent for each behavioral category were measured using event-recorder software on a personal computer.

**Statistical analysis.**   As male intromission has frequently been confused with female lordosis (receptivity index described in Inoue, *et al.* [31]), this experiment aimed to clearly illustrate the distinctions between IR and LQ in the bilevel chamber. We compared %Intro (IR × 100) and LQ by paired Student's *t*-test (effect size test: Cohen's *d* test).

## Results of Experiment I

Fig 2 shows the estimated LQ based on males' intromission in the ordinary cage test (last trial of mating experience sessions), and %Intro and LQ obtained by accurate observation in the bilevel chamber test. The mean LQ in the bilevel chamber test was higher than that in the ordinary cage test; however, statistical comparison is not possible due to multiple differences, such as test conditions, sexual experience, behavioral parameters, etc. On the other hand, there were significant differences between %Intro (used to estimate LQs in the ordinary cage test) and the actual LQ measured in the bilevel chamber test ($t = 15.93$, $df = 7$, $p < 0.001$, $d = 4.13$). Quantifying the incidence of lordosis to males' mounts and intromissions separately revealed that females exhibited lordosis in response to almost every intromission (99.1%, but not 100%) and partly to males' mounts (55.8%).

In the bilevel chamber test, we could record females' rejection-like behavior, something that cannot be observed in the ordinary cage test. Experimental females showed 15.1 ± 5.51 (mean ± SEM) rejection-like behaviors during a bilevel chamber test trial.

## Discussion of Experiment I

In this first experiment, we attempted to accurately evaluate female sexual behavior using a newly devised bilevel apparatus. As mentioned above, mouse lordosis does not have the conspicuous behavioral characteristics. In fact, %Intro, used for the estimation of LQ in the ordinary cage test, was significantly lower than the LQ based on the accurate observation in the bilevel chamber test. Thus, although lordosis accompanies almost every male's intromission as having been thought, the difference in LQs between the ordinary cage and the bilevel chamber tests was obtained because estrous female mice showed lordosis in response to not only intromissions but also mounts that did not involve the penile intromission, indicating that male's intromission is not necessarily required for lordosis.

In addition, the bilevel chamber test could quantify rejection-like behavior, which could not be detected as a specific behavioral pattern in the ordinary cage test. This behavior is

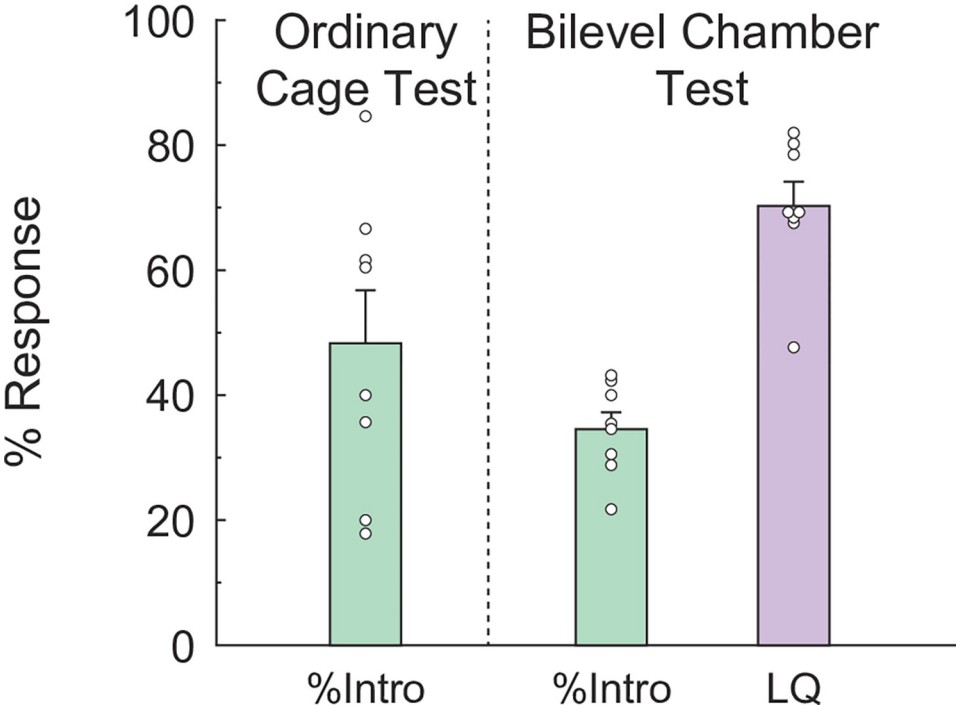

**Fig 2. Comparison of lordosis measurements between tests in an ordinary cage and in the new bilevel apparatus.**
LQs in the ordinary test cage (left column) were estimated by males' intromission. LQs in the bilevel apparatus (right column) were based on accurate observation (see methods). The accurate LQs significantly differed from the number of intromissions (middle column), for which the LQ estimation in the ordinary cage test was used. Small circles indicate raw data, and the vertical bar indicates the SEM. ***$p < 0.001$.

clearly distinct from what is commonly referred to as so-called aggressive behavior, but is rather classified as agonistic behavior [32]. In most cases of rejection-like behavior, females reared up and extended their forepaws or hindlimbs to prevent male approaches without any consequent attacks. The rejection-like behavior means that the female does not accept males' mount, so it can be considered to be the opposite expression of female proceptivity. This can be an important measure for the study of sexual behavior in female mice, since the conventional tests lack a definite parameter for proceptivity, such as ear-wiggling or hopping and darting observed in female rat sexual behavior.

## Experiment II: Effect of AVP receptor deficiency

### Materials and methods

**Subjects.**   Wild type (WT, n = 6), aKO (n = 6), bKO (n = 7), and dKO (n = 5) sexually mature female mice (C57BL/6 background, 10–13 weeks old) from a breeding colony in the National Center for Child Health and Development, Japan (NCCHD, Tokyo, Japan) were used [33, 34]. They were ovariectomized under isoflurane anesthesia >1 week before starting the behavioral tests. Each genotypic group (2–4 mice per cage) was maintained under the same controlled environment described in Experiment 1. Stimulus males for sexual behavior tests with previous mating experience with another set of female mice (not experimental subjects) were used as well. In the Experiments II and III, the males that accomplished ejaculation at least twice during the screenings were used as stimulus males.

**Sexual behavior test.** As in Experiment I, all subject females had a 30 min mating experience once in the ordinary cage. Then, they were subjected twice to the 20 min bilevel chamber test with 1-week interval. The pairing of subjects and stimulus males was arranged to avoid encountering the same males throughout these tests.

**Statistical analysis.** LQs, durations of lordosis and frequency of rejection-like behavior in the genotypes (WT, aKO, bKO, and dKO) × 2 tests were examined by two-way ANOVA with repeated measures (effect size test: partial Eta squared), followed by multiple comparisons with Bonferroni tests (effect size test: Cohen's *d* test).

## Results of Experiment II

**Lordosis.** Fig 3A shows the mean LQs in each genotypic group in bilevel chamber tests. Two-way ANOVA indicated a significant main effect of genotypes ($F_{3,20} = 11.80$, $p < 0.001$,

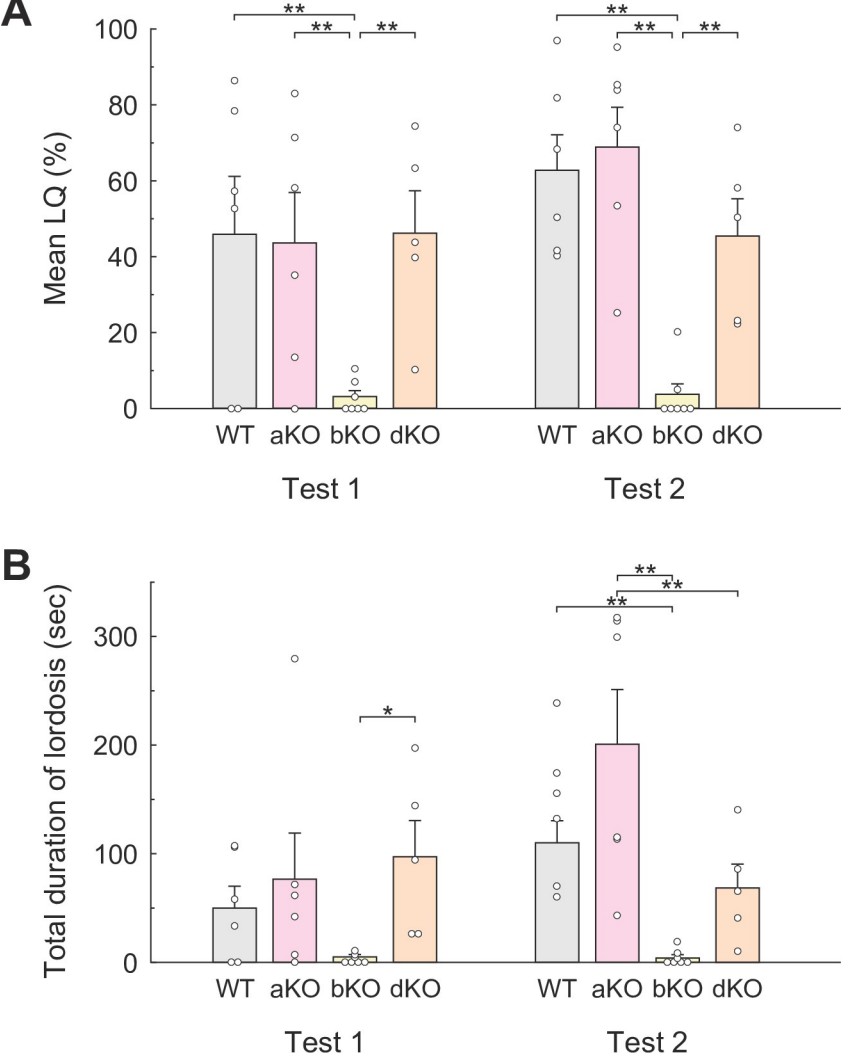

**Fig 3. Lordosis behavior in each genotypic group during the bilevel chamber test in Experiment II. (A)** Mean lordosis quotients [LQ = (number of lordosis / total number of mounts and intromissions received)×100]. **(B)** Total duration of each lordosis response (sec). Small circles indicate raw data, and vertical bars indicate the SEM. *$p < 0.05$, **$p < 0.01$.

partial $\eta^2 = 0.64$). Bonferroni *post hoc* tests revealed significant differences between the LQs of WT females (Test 1, 46.0 ± 15.44; Test 2, 63.1 ± 9.46) and bKO females (Test 1, 3.0 ± 1.60; Test 2, 3.6 ± 2.83)($t = 4.51^{**}$, $d = 1.67$ and $t = 6.25^{**}$, $d = 3.59$, $^{**}p < 0.01$, respectively), but not of aKO females (Test 1, 43.7 ± 13.44; Test 2, 69.2 ± 10.55). Interestingly, the LQs of dKO females in both tests (Test 1, 46.5 ± 11.00; Test 2, 45.5 ± 10.02) were comparative to those of WT females and significantly higher than those of bKO females in both tests ($t = 4.33^{**}$, $d = 2.73$ and $t = 4.18^{**}$, $d = 2.53$, $^{**}p < 0.01$, respectively).

Fig 3B presents another lordosis parameter, total lordosis duration in each test. Two-way ANOVA again showed a significant main effect of genotypes ($F_{3,20} = 10.83$, $p < 0.001$, partial $\eta^2 = 0.62$). Bonferroni *post hoc* tests indicate significant increase of the lordosis duration in Test 1 of dKO females compared to bKO females ($t = 3.27$, $p < 0.05$, $d = 1.95$). Although no statistical significance was yielded, the lordosis duration in bKO females was almost suppressed in Test 1. In Test 2, the lordosis duration was significantly suppressed in bKO females compared to WT females ($t = 3.95$, $p < 0.01$, $d = 3.12$) and in dKO females compared to aKO females ($t = 4.54$, $p < 0.001$, $d = 1.36$).

**Rejection-like behavior.** An ANOVA indicated significant main effects of both genotypes (between subjects) and test repetitions (within subjects) ($F_{3,20} = 3.43$, $p < 0.05$, partial $\eta^2 = 0.34$ and $F_{1,20} = 26.74$, $p < 0.001$, partial $\eta^2 = 0.57$, respectively). Subsequent *post hoc* analyses by Bonferroni test showed a significantly lower number of rejection-like behaviors in aKO females than in WT females ($t = 3.94$, $p < 0.01$, $d = 2.64$) and in aKO females than bKO females ($t = 4.93$, $p < 0.001$, $d = 1.39$); as did dKO females in comparison to bKO counterparts ($t = 3.45$, $p < 0.05$, $d = 0.92$), demonstrating the inhibitory effect of a deficient v1a receptor on rejection-like behaviors (Fig 4A). This analysis also revealed that test repetition significantly decreased the number of rejection-like behaviors in all groups (WT, $t = 5.86^{**}$, $d = 1.49$; aKO, $t = 2.22^*$, $d = 0.87$; bKO, $t = 8.96^{**}$, $d = 0.95$; dKO, $t = 3.80^{**}$, $d = 0.83$, $^*p < 0.05$, $^{**}p < 0.01$), resulting in no significant differences among the groups in Test 2.

**Sharing the floor with stimulus males.** Fig 4B presents mean % time shared on the same floor with stimulus males by females of each genotypic group. No significant difference was observed between any groups.

## Experiment III: Effect of oxytocin receptor deficiency

### Materials and methods

Sexually mature OTRKO female mice (C57BL/6 background, 10–13 weeks old; n = 12) belonged to our breeding colony, originally supplied from Tohoku University (Sendai, Japan). The *otr* gene was displaced by the *venus* gene derived from *Aequorea coerulescens*; thus, *venus*$^{+/+}$ is equal to *otr*$^{-/-}$ [35]. WT females were also prepared from their littermate (n = 9). Housing conditions were the same as in Experiment II as were the process of stimulus male preparation and behavioral tests.

### Result of Experiment III

**Lordosis.** A two-way ANOVA for LQs revealed no significant main or interaction effects (genotypes $F_{1,19} < 1$; repetition $F_{1,19} = 3.90$, $p = 0.063$, partial $\eta^2 = 0.17$; interaction $F_{1,19} < 1$, Fig 5A). Similarly, ANOVA for the total duration of lordosis showed no significant main effect of genotypes ($F_{1,19} = 2.05$, *ns*) but a significant main effect of test repetition ($F_{1,19} = 16.72$, $p < 0.001$, partial $\eta^2 = 0.47$, Fig 5B).

**Rejection-like behavior.** A two-way ANOVA (Fig 6A) followed by Bonferroni *post hoc* analysis indicating significance of the main effects of both genotypes ($F_{1,19} = 7.68$, $p = 0.05$, partial $\eta^2 = 0.29$) and test repetition ($F_{1,19} = 5.63$, $p < 0.05$, partial $\eta^2 = 0.23$); thus OTRKO

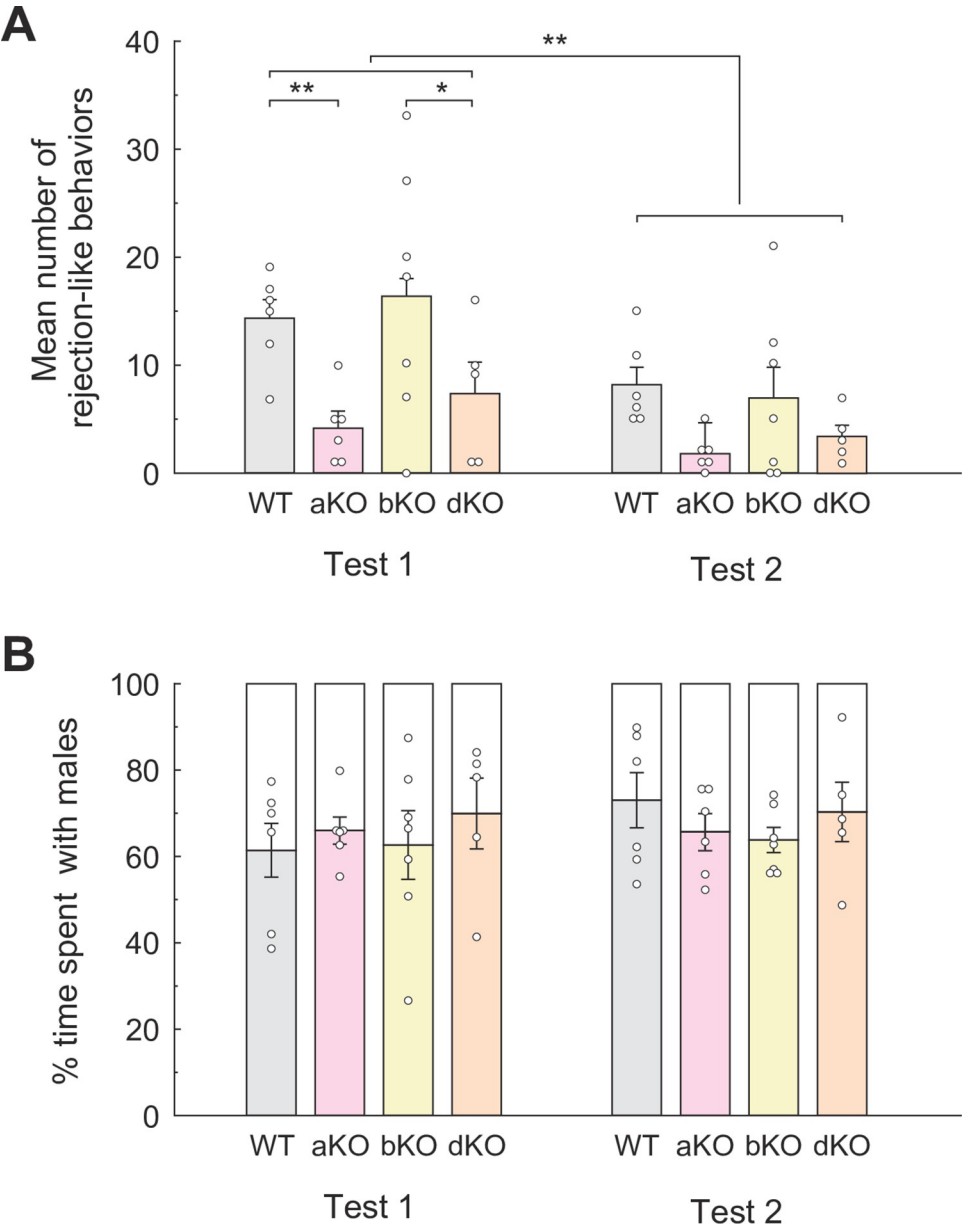

**Fig 4. Rejection-like behavior and % time sharing the same floor with males in each genotypic group during the bilevel chamber test in Experiment II. (A)** Mean number of rejection-like behaviors of each genotypic group in the bilevel chamber test in Experiment II. **(B)** Mean % time spent on the same floor with males for each genotypic group. Small circles indicate raw data, and a vertical bars indicate the SEM. $^*p < 0.05$, $^{**}p < 0.01$.

females displayed more rejection-like behaviors than WT females in both Tests 1 ($t = 3.46$, $p < 0.01$, $d = 0.92$) and 2 ($t = 3.24$, $p < 0.01$, $d = 1.26$; Fig 6A). The effect of test repetition was examined by *post hoc* Bonferroni tests, which showed that significantly decreased rejection-like behavior in Test 2 compared to Test 1 in both WT ($t = 2.25$, $p < 0.05$, $d = 0.68$) and OTRKO ($t = 2.55$, $p < 0.05$, $d = 0.50$) females.

**Sharing the floor with stimulus males.**   Fig 6B shows the mean % time shared on the same floor with stimulus males by females of each genotypic group. Interestingly, a significant main effect of genotypes was observed in two-way ANOVA ($F_{1,19} = 10.90$, $p < 0.01$, partial $\eta^2$

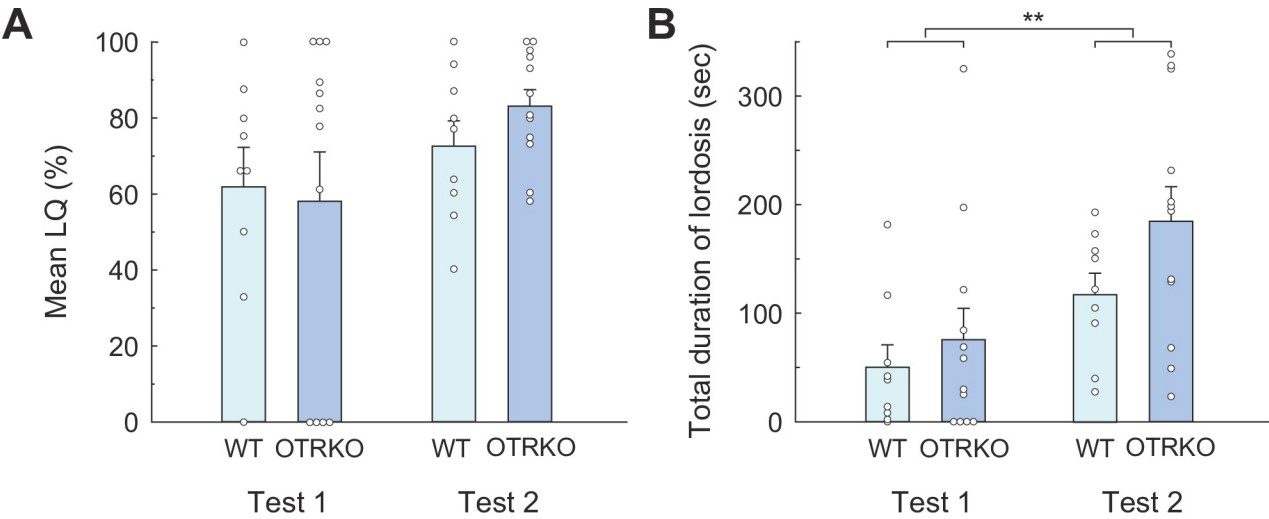

**Fig 5. Lordosis behavior of each genotypic group in the bilevel chamber test in Experiment III. (A)** Mean LQ and **(B)** mean duration of each lordosis response (sec). Small circles indicate raw data, and vertical bars indicate the SEM. **$p < 0.01$.

= 0.36). Further, *post hoc* Bonferroni test indicated significance in both Test 1 ($t = 2.49$, $p < 0.05$, $d = 0.60$) and Test 2 ($t = 4.57$, $p < 0.01$, $d = 2.45$), such that OTRKO females shared the same floor with stimulus males significantly longer than WT females.

## Discussion of Experiments II and III

Using our novel bilevel apparatus for female mice, we examined the effect of gene deficiency in AVP or OT receptors on female sexual behavior. Experiment II demonstrated that aKO females displayed a normal level of lordosis compared with WT females whereas bKO females showed almost no lordosis, having significantly lower LQs than those of WT females. Conversely, dKO females showed a comparable level of lordosis with WT and aKO females. This

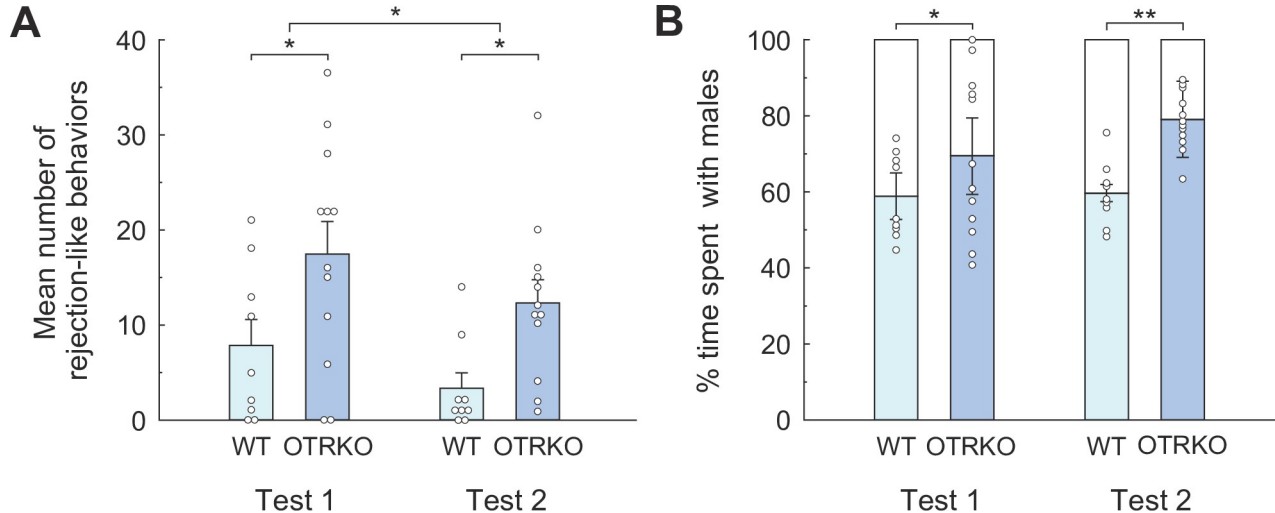

**Fig 6. Rejection-like behavior and % time sharing the same floor with males in each genotypic group during the bilevel chamber test in Experiment III. (A)** Mean number of rejection-like behaviors of each genotypic group in the bilevel chamber test in Experiment III. **(B)** Mean % time spent on the same floor with males for each genotypic group. Small circles indicate raw data, and vertical bars indicate the SEM. *$p < 0.05$, **$p < 0.01$.

finding indicates that, despite its important role in the expression of lordosis, the v1b receptor is not essential. In addition, although no difference was observed between aKO and WT females, the result on dKO strongly suggests a potentially facilitatory effect of aKO on lordosis.

One possible explanation for these phenomena is that neurons with the v1a receptor latently suppress lordosis, while neurons with the v1b receptor strongly promote lordosis. Consequently, the exclusive deletion of the v1a receptor has no effect, whereas the exclusive deletion of the v1b receptor seriously suppresses lordosis. The role of the v1a receptor is unveiled when both the v1a and v1b receptors are deleted. Another plausible explanation is that neurons with the v1a receptor may suppress lordosis but are constantly inhibited by neurons with the v1b receptor. According to this model, the exclusive deletion of the v1a receptor has no impact because these neurons are already inactivated by neurons with v1b receptor. Conversely, the exclusive deletion of the v1b receptor releases the lordosis inhibition by v1a receptors, leading to suppression of lordosis. The double KO of the v1a and v1b receptors nullifies this system, resulting in normal levels of lordosis. In any case, it is imperative to assume the existence of another neural circuit indispensable for lordosis. The lordosis circuit may be regulated by the balance of the actions between these two types of AVP receptors.

Such opposing roles of v1a and v1b receptors in lordosis may have resulted in contradictory results concerning the promotive [7] and suppressive [22] effects of AVP administration on rat lordosis. Although v1a receptor binding to the dorsal part of the LS was significantly higher in females than in males [30], v1a receptors in the LS suppressively regulate aggression [36] and social play behavior in female rats [37, 38]. Moreover, the LS is a well-known part of the inhibitory system for lordosis in female rats [29], suggesting that in female mice the LS may inhibit lordosis. Furthermore, a dependency of AVP immunoreactive cells and fibers in LS to sex hormones has been previously reported [39]. In estrous females, estrogen and progesterone might act on LS neurons to disinhibit the expression of lordosis [40, 41]. Thus, AVP and v1a receptors are proposed to be involved in lordosis's disinhibition (facilitation).

Alternatively, the deficiency of v1b receptors decreases ultrasonic distress calls in neonatal mice [42], increases aggressive behavior in male mice [43, 44], and eliminates male sexual preference toward estrus odor [43]. However, our knowledge on the brain function of v1b receptors is lower than that of v1a receptors. The v1b receptors have been found in the olfactory bulb, supraoptic, suprachiasmatic, and dorsomedial hypothalamic nuclei, including the piriform/entorhinal cortices, hippocampus, substantia nigra, and dorsal motor nucleus of the vagus in the rat brain [45]. Further studies are required to clarify the involvement of these regions in the regulation of lordosis in mice.

In Experiment III, OTRKO females showed comparable levels of lordosis to that of WT females. As discussed below, even though OTRKO females display increased rejection-like behavior, if males eventually mount properly, OTRKO females can exhibit lordosis. In contrast to previous studies in female rats where intraventricular OT injection with OT or OT antagonist promotes [6–8], or inhibits lordosis [8], respectively, our results indicated no direct OTR involvement in mouse lordosis. A study reported that OT (ligand) knockout suppressed lordosis in female mice [19]. Together with our results, it suggests the possible involvement of a cross-talk reaction of OT to v1b receptors in mouse lordosis.

On the other hand, rejection-like behavior significantly increased in OTRKO females compared to WT females. That is, OTRKO females sufficiently showed lordosis in response to male mounts regardless of their refusal to approach or contact males. Whereas rejection-like behavior was not affected in bKO females, rejection-like behavior in both aKO and dKO females was significantly decreased. Furthermore, rejection-like behavior may be influenced by sexual experience. Across all groups, rejection-like behavior was significantly reduced in

Test 2 compared to Test 1. While sexual experience is known to facilitate sexual behavior, it appears that the receptors of AVP and OT may not be involved in this promoting effect.

Such reduction of rejection-like behaviors may be reflected in proceptivity known in female rat sexual behavior. In fact, OT action promotes not only lordosis but also soliciting behavior, such as hopping and darting, while suppressing rejection-like behaviors, such as kicking males, in female rats [46]. Interestingly, OTRKO females shared the floor with stimulus males significantly longer than WT females, which we do not think is due to changes in pacing behavior in OTRKO female mice. It is likely that OTRKO females spent longer time engaging in rejection-like behavior than WT females, resulting in more time on the same floor with stimulus males.

Proceptive behaviors in female rats are strongly enhanced by progesterone [47, 48]. This implies that OT's facilitatory effects of female sexual behavior require progesterone [46, 49]. Further, the facilitatory effects of progesterone or OT treatments in estrogen-primed rats are reported to be mediated in the midbrain ventral tegmental area [50] and the ventromedial hypothalamus [51, 52]. Accordingly, those neural mechanisms may also be involved in rejection-like behavior in female mice, which have no conspicuous soliciting behaviors.

Currently, research on sexual behavior in female mice is still in its infancy. In the future, it will be necessary to further clarify the relationships and interactions between OT and AVP (along with their receptors) and sex steroids in the neural regulation of female mouse sexual behavior.

## Conclusion

By developing an improved observation method for sexual behavior in female mice, we could distinguish two elements of sexual behavior: facilitation of lordosis and reduction of rejection-like behavior, reflecting receptivity and proceptivity, respectively. Using this new method, we investigated the involvement of AVP and OT receptors in female mouse sexual behavior. Based on our results, the v1a receptor potentially inhibits receptivity and proceptivity, whereas the v1b receptor plays an important, but not essential, role in receptivity. Moreover, OTR potentiates proceptivity by suppressing rejection-like behavior.

## Supporting information

**S1 Fig. Mean number of stimulus male mounts to females of each genotype group during 20 min observations in Experiments II and III.** No significant difference was observed between any two genotype groups or tests.
(PDF)

**S2 Fig. Mean intromission ratios (%, number of intromissions/total number of mounts • 100) shown by stimulus males to females of each genotype group in each observation in Experiments II and III.** Results were statistically analyzed by ANOVA, followed by Bonferroni, $^*p < 0.05$: significant difference between bKO and dKO ($F_{1,40} = 3.05$), $^{**}p < 0.01$: significant difference between WT and bKO ($F_{1,40} = 3.80$)/bKO and dKO ($F_{1,40} = 3.88$), and $^{##}p < 0.01$: significant difference between Tests 1 and 2 ($F_{1,14} = 4.19$).
(PDF)

**S3 Fig. Intromission and ejaculation latencies of stimulus males to females of the AVP receptor genotype groups during 20 min observations in Experiment II (shown by reversed survival curves).** Survival analysis revealed significant differences, indicated by asterisks. $^*p < 0.05$, $^{**}p < 0.01$.
(PDF)

**S4 Fig. Intromission and ejaculation latencies of stimulus males to WT and OTRKO females during 20 min observations in Experiment III (shown by reversed survival curves).** Survival analysis revealed no significant difference between WT and OTRKO. (PDF)

## Author Contributions

**Conceptualization:** Himeka Hayashi, Yasuhiko Kondo.

**Formal analysis:** Himeka Hayashi, Yasuhiko Kondo.

**Funding acquisition:** Yasuhiko Kondo.

**Investigation:** Himeka Hayashi.

**Methodology:** Himeka Hayashi, Yasuhiko Kondo.

**Project administration:** Yasuhiko Kondo.

**Resources:** Kie Shimizu, Kazuaki Nakamura, Katsuhiko Nishimori.

**Supervision:** Yasuhiko Kondo.

**Writing – original draft:** Himeka Hayashi, Yasuhiko Kondo.

**Writing – review & editing:** Yasuhiko Kondo.

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
