## [Decision Letter · Decision Letter 0]

23 Jan 2024

PONE-D-23-41361The bilevel chamber revealed differential involvement of vasopressin and oxytocin receptors in female mouse sexual behaviorPLOS ONE

Dear Dr. Kondo,

Thank you for submitting your manuscript to PLOS ONE. After careful consideration, we feel that it has merit but does not fully meet PLOS ONE’s publication criteria as it currently stands. Therefore, we invite you to submit a revised version of the manuscript that addresses the points raised during the review process. The manuscript has been assessed by two experts in the field; in particular, I would advise you to pay attention to the comments from Reviewer 1. Please find the detailed critiques in the reviews below.

We look forward to receiving your revised manuscript.

Kind regards,

Nirakar Sahoo, PhD

Academic Editor

PLOS ONE

5. We note that Figure 1 in your submission contain copyrighted images. All PLOS content is published under the Creative Commons Attribution License (CC BY 4.0), which means that the manuscript, images, and Supporting Information files will be freely available online, and any third party is permitted to access, download, copy, distribute, and use these materials in any way, even commercially, with proper attribution. For more information, see our copyright guidelines: http://journals.plos.org/plosone/s/licenses-and-copyright.

Reviewers' comments:

Reviewer's Responses to Questions

**Comments to the Author**

1. Is the manuscript technically sound, and do the data support the conclusions?

Reviewer #1: Yes

Reviewer #2: Yes

2. Has the statistical analysis been performed appropriately and rigorously? 

Reviewer #1: I Don't Know

Reviewer #2: Yes

3. Have the authors made all data underlying the findings in their manuscript fully available?

Reviewer #1: Yes

Reviewer #2: Yes

4. Is the manuscript presented in an intelligible fashion and written in standard English?

Reviewer #1: Yes

Reviewer #2: Yes

5. Review Comments to the Author

Reviewer #1: In the research article entitled “The bilevel chamber revealed differential involvement of vasopressin and oxytocin receptors in female mouse sexual behavior”, the authors attempted to measure the female mice sexual behavior by constructing a new apparatus “bilevel chamber”. Also, they evaluated the effects of the arginine vasopressin (AVP) receptors v1a and v1b, and the oxytocin (OT) receptors on female mice sexual behavior in mice, by making the corresponding knockout female mice. Authors took the advantage of the easy genetic manipulation in mice to overcome the limitations (in comparison to rats) in studying the female mice sexual behavior. However, there are some concerns observed and the following minor and major issues listed.

Comments:

1) page 2:

a) line 10: The aKO females showed decreased rejection-like behavior, based on the discussion on page 20, lines 360-361, and Fig. 4A. Replace “increased” with “decreased”.

2) page 3:

a) line 22: Replace “evolutionary” with “evolutionarily”.

b) lines 31-33: Cite the references 9 and 10 after the word MPOA; or Cite all the references at the end of the sentence.

3) page 4:

a) line 40: The word “typical” is repeated twice. It may be corrected to one of the following:

“a typical diagnostic behavior” (OR) “a typical diagnostic type of behavior”.

b) line 47: Replace “Meanwhile” with “Also, it was reported/discovered that”.

c) line 51: It’s good to provide a figure for showing the sequence and structural similarity between AVP and OT. Otherwise give a reference.

d) lines 53-54: References should be given for lines 53 and 54.

4) page 5:

a) line 70: Use either “the” or “a”. Don’t use both, one after another.

b) line 74: The word “behavior” is repeated twice. This word maybe deleted at the end of the sentence.

5) page 6:

a) line 81: Give the “n” value.

b) lines 89-90: Give the brand/trade names for estradiol and progesterone used in this study.

c) line 89: Correct the “estradiol-17β benzoate” name appropriately to either “17β-estradiol 3-benzoate (EB)” or “17β-estradiol-3-benzoate (EB)”, in order to avoid confusion with the other estradiol benzoate variant “estradiol 17β-benzoate (E2-17B)”.

d) line 96: Replace “be adapt” with “get adapted”.

6) page 7:

a) line 98: Continue the sentence as “…………30 min, whichever is the first.”, if it’s appropriate.

b) lines 101-102: Give the LQ formula in a proper mathematical formula/equation format. Also, if you want to give this in the text, please express this correctly to “A lordosis quotient [LQ, (number of lordosis / total number of mounts and intromissions received) × 100] was calculated………….”.

c) line 103: Modify the sentence to “………..repeated weekly until an LQ ≥ 50% was observed.”.

d) How many weeks did it take to observe an LQ ≥ 50%. It’s better if it’s mentioned here.

e) lines 110-112: How the constrained sideways, due to the narrow depth of the bilevel chamber, help the experimenter better evaluating the female lordosis. Is it by physically forcing/making the female mice body to squeeze as lordosis and/or rise its tail to better heights, due to space constraint. It’s good if it’s explained better.

Also, since it’s an external/instrumental physical/mechanical constraint on the mice body and its free movement, is it considered as disadvantageous for studying the true behavior of the mice. This constraint may act as a stimulus, similar to the male mount, and cause/create a false lordosis. I said false lordosis since it’s not because of the true stimulus (male mount). Is this a con/disadvantage of the newly designed apparatus (bilevel chamber).

7) page 8:

a) line 123: Use “the” after the word “Both”.

b) line 126: Replace “Then, when” with “While”, or with appropriate word.

c) line 128: Continue the sentence as “…………male ejaculation, whichever is the first.”, if it’s appropriate.

d) Why the ordinary cage test was carried out for 30 min (page 7, line 98) and the bilevel chamber test for 20 min (page 8, line 128).

8) page 9:

a) line 138: Remove “were obtained”, if it’s appropriate.

b) line 141: Use “the” after the word “classified”.

c) line 146: Is it possible for intromission event to happen without the lordosis event. How frequently this bout (Intromission without lordosis) was observed. Does this signifies that lordosis is not a necessary event to occur during mice mating/intercourse.

d) line 148: What is the minimum, maximum, average length of different bouts/events explained in this page. How could you measure the events with “> 2 sec” precision.

Also, how did you monitor and count/measure the no. of lordosis events in female mice; and the no. of mount, intromission, and ejaculation events in male mice. If it’s manual monitoring, how accurate is the counting.

e) lines 152-153: Give the IR formula in a proper mathematical formula/equation format. Refer the comment 6b (page 7).

9) page 10:

a) line 156: If the statement “IR was considered equally to LQ” was from the previously published literature, cite the suitable reference for this. And if this is your own observation/statement, explain why you considered this.

b) line 157: Correct appropriately to “……..LQ, as clearly distinguished…….”, if needed.

c) lines 161-163: Why the female lordosis events, and so the true LQ, could not be observed/measured accurately in the ordinary cage. How differently the bilevel chamber help the experimenter to observe the female lordosis events accurately. Just to mention here, the male mount, intromission, and ejaculation events are observed equally well in both the ordinary cage and bilevel chamber apparatus.

And, if the true LQ cannot be measured in the ordinary cage test, why do you consider the “estimated LQ”, measured based on the IR and %Intro, to compare with the true LQ and %Intro calculated in the bilevel chamber test. Why don’t you use the “%Intro” in place of “estimated LQ”.

d) lines 168-169: Correct the sentence appropriately to “……….and intromissions separately revealed that females exhibited………….”.

e) line 172: Replace “behavior” appropriately with “behavioral”, if needed.

10) page 11:

a) lines 180-181: Why the female rejection-like behavior events could not be observed/measured accurately in the ordinary cage. How differently the bilevel chamber help the experimenter to observe the female rejection-like behavior events accurately.

b) line 182: Remove the word “in”.

11) page 12:

a) line 210: Use “and” before the word “dKO”.

b) line 210: Why is the “n” value difference between the WT, aKO, bKO, and dKO mice groups.

12) page 13:

a) line 223: Does this line means that different stimulus males were used in the subsequent/different bilevel chamber tests. Make it clear.

13) page 14:

a) lines 244-245: The LQ formula/equation on page 14, lines 244-245 is different from that on page 7, lines 101-102. Correct this appropriately.

b) line 245: Replace “s” with “sec”, for seconds.

c) Correct the following:

line 245: Mean duration

line 248: Total duration

Fig. 3B: Total duration

14) page 15:

a) lines 251-253: Explain it better. For example, in the test 1 also the lordosis duration was suppressed in bKO females compared to WT females.

b) line 259: Correct appropriately to “…post hoc analyses by Bonferroni test showed…….”.

c) line 269: Figure title should be given in the Fig. 4 legend.

15) page 16:

a) Correct the following:

line 270: Mean % time

line 275: Time

Fig. 4B: % time

b) Lines 280, 283: Why is the “n” value difference between the WT, OTRKO, aKO, bKO, and dKO mice groups.

16) page 17:

a) line 296: Replace “s” with “sec”, for seconds.

b) line 304: How do you explain the effect of test replication.

17) page 18:

a) line 308: Figure title should be given in the Fig. 6 legend.

b) Correct the following:

line 309: Mean % time

line 314: Time

Fig. 6B: % time

c) What is the significance of % time spent with males, in the context of female sexual behavior.

18) page 19:

a) line 328: If the v1b receptor is not essential for lordosis expression, then why bKO showed almost no lordosis. Interpret this in a better way.

b) line 330: “in dKO females, aKO restored lordosis from the suppression by bKO up to the WT level”. How do you justify this hypothesis. Any experimental evidence. And how the dKO showed normal levels of lordosis, while the bKO showed no lordosis. It means there may be elements (other than v1a and v1b receptors) involving in nullifying/inhibiting the effect of bKO in presence of aKO. Can you shed light on this. If you propose that the crosstalk between OT and v1b receptors (page 20, line 356) maybe the underlying cause for this, please explain it better. And if so, how did you propose this hypothesis, and any attempts made for experimental evidence. Any other elements may involve in this ?.

19) page 21:

a) How the OTRKO females exhibit normal lordosis levels, while showing increased rejection-like behavior. What is the stimulus for lordosis in this case, if it’s not the male mount (which is not possible due to rejection by female mice).

b) lines 378-379: Does the “rejection-like behavior” reflect “proceptivity”. Please explain this.

20) Have you attempted to evaluate the effects of AVP, OT, AVP/OT combination, progesterone administrations/treatments in WT, aKO, bKO, dKO, OTRKO mice, similar to the rat lordosis experiments (page 19, lines 333-334; page 20, lines 352-353; page 21, lines 371).

21) Did you monitor and measure the Intromission and ejaculation events in experiments II and III.

22) Figure 2:

a) Replace “Ordinary” with “Ordinary Cage Test”, if appropriate

b) Replace “Bilevel Chamber” with “Bilevel Chamber Test”, if appropriate

23) P values and the corresponding bars should be given in the following figures.

a) Figure 3: Fig. 3B panel: P values and the corresponding bars for all the groups in Test 1.

b) Figure 4: Fig. 4B panel: P values and the corresponding bars for all the groups in Test 1 and 2.

c) Figure 5: Fig. 5A panel: P values and the corresponding bars for all the groups in Test 1 and 2.

Reviewer #2: Authors in their manuscript described a new test for sexual behavior in mice and the role of vasopressin and oxytocin receptors in mice female sexual behavior. While the test itself seems interesting, new findings in sexual behavior can be a good addition to our knowledge of how such social neuropeptides as AVP and OT function. Here are several questions to be asked before considering the manuscript for publication.

1) Figure 3A and b WT upper significance bar misleading, it looks like WT mice have statistical differences with both aKO and bKO, while the difference is only with bKO. Please check and modify the significance bars.

2) While the results of rejection-like behavior are clear, the influence of V1A and V1B receptors on lordosis seems puzzling. May you address more in the discussion why the absence of both receptors has the same results as their presence? While it seems to be some kind of interaction, it’s still unclear how knockout of V1AR can recover the effects of knockout of V1BR.

3) Why you choose a different strain for experiment I? (C57BL6 for II and III is clear to me, because KO mice are based on the same strain)

4) You used the terms receptivity and proceptivity in the conclusion and briefly used them in the discussion; May you clearly state in the introduction what you mean by those terms?

5) May you indicate in your methods part how you clean the Believel chamber at the end of the test in order to remove odor influence between tests?

6. PLOS authors have the option to publish the peer review history of their article (what does this mean?). If published, this will include your full peer review and any attached files.

Reviewer #1: No

Reviewer #2: No

---

## [Author Response · Author response to Decision Letter 0]

19 Mar 2024

To Reviewer 1:

Thank you sincerely for your thoughtful and comprehensive review. We have thoroughly examined your comments and made revision to our MS accordingly. We believe that, thanks to your valuable advice, our MS has greatly improved. We trust that our corrections meet your criterion.

Below, we describe the changes we made to our MS following your comments. (an attached file contains changed points highlighted, each indicated your comment No.)

1) page 2:

a) line 10: The aKO females showed decreased rejection-like behavior, based on the discussion

on page 20, lines 360-361, and Fig. 4A. Replace “increased” with “decreased”.

Line 10: replaced it.

2) page 3:

a) line 22: Replace “evolutionary” with “evolutionarily”.

Line 10: replaced it.

b) lines 31-33: Cite the references 9 and 10 after the word MPOA; or Cite all the references at the

end of the sentence.

Line 32: The citation point was moved.

3) page 4:

a) line 40: The word “typical” is repeated twice. It may be corrected to one of the following:

“a typical diagnostic behavior” (OR) “a typical diagnostic type of behavior”.

Line 40: Thank you for pointing out our mistake. removed one.

b) line 47: Replace “Meanwhile” with “Also, it was reported/discovered that”.

Line 52: replaced it.

c) line 51: It’s good to provide a figure for showing the sequence and structural similarity between

AVP and OT. Otherwise give a reference. 

Thank you for your advice. However, so ref#20 we cited contains a figure describing the sequences, that we did not think a similar figure is necessary in our MS.

d) lines 53-54: References should be given for lines 53 and 54.

Line 59: added citations.

4) page 5:

a) line 70: Use either “the” or “a”. Don’t use both, one after another.

Line 71: replaced it.

b) line 74: The word “behavior” is repeated twice. This word maybe deleted at the end of the

sentence.

Line 81: removed it.

5) page 6:

a) line 81: Give the “n” value.

Line 87: added “n” value.

b) lines 89-90: Give the brand/trade names for estradiol and progesterone used in this study.

Lines 96-97: added the brand name.

c) line 89: Correct the “estradiol-17β benzoate” name appropriately to either “17β-estradiol 3-

benzoate (EB)” or “17β-estradiol-3-benzoate (EB)”, in order to avoid confusion with the other

estradiol benzoate variant “estradiol 17β-benzoate (E2-17B)”.

Line 96: corrected it.

d) line 96: Replace “be adapt” with “get adapted”.

Line 103: changed it.

6) page 7:

a) line 98: Continue the sentence as “…………30 min, whichever is the first.”, if it’s appropriate.

Line 105: added your clause.

b) lines 101-102: Give the LQ formula in a proper mathematical formula/equation format. Also,

if you want to give this in the text, please express this correctly to “A lordosis quotient [LQ,

(number of lordosis / total number of mounts and intromissions received) × 100] was

calculated………….”.

Line 108: rewrote the equation.

c) line 103: Modify the sentence to “………..repeated weekly until an LQ ≥ 50% was observed.”.

Line 110: modified it following you.

d) How many weeks did it take to observe an LQ ≥ 50%. It’s better if it’s mentioned here.

Line 111: added the information.

e) lines 110-112: How the constrained sideways, due to the narrow depth of the bilevel chamber,

help the experimenter better evaluating the female lordosis. Is it by physically forcing/making the

female mice body to squeeze as lordosis and/or rise its tail to better heights, due to space constraint.

It’s good if it’s explained better.

Also, since it’s an external/instrumental physical/mechanical constraint on the mice body and its

free movement, is it considered as disadvantageous for studying the true behavior of the mice. This

constraint may act as a stimulus, similar to the male mount, and cause/create a false lordosis. I said

false lordosis since it’s not because of the true stimulus (male mount). Is this a con/disadvantage

of the newly designed apparatus (bilevel chamber).

Lines 118-121: we modified and added the description of the bilevel chamber. The depth of the apparatus constrained the body direction but did not prevent any behavior they showed.

7) page 8:

a) line 123: Use “the” after the word “Both”.

Line 136: inserted it.

b) line 126: Replace “Then, when” with “While”, or with appropriate word.

Line 139: replaced “Then, when" with "Subsequently,”

c) line 128: Continue the sentence as “…………male ejaculation, whichever is the first.”, if it’s

appropriate.

Line 141: added your clause.

d) Why the ordinary cage test was carried out for 30 min (page 7, line 98) and the bilevel chamber

test for 20 min (page 8, line 128).

In the initial phase of this study, we conducted sexual behavior tests for 30 min in the bilevel chamber. However, through several pilot tests, we determined that a 20-min test is sufficient to measure sexual behavior. Consequently, we have settled on this duration thereafter.

8) page 9:

a) line 138: Remove “were obtained”, if it’s appropriate.

Line 154: changed it.

b) line 141: Use “the” after the word “classified”.

Line 157: added “the”.

c) line 146: Is it possible for intromission event to happen without the lordosis event. How

frequently this bout (Intromission without lordosis) was observed. Does this signifies that lordosis

is not a necessary event to occur during mice mating/intercourse. 

Lines 187-189: these are described in the Result.

d) line 148: What is the minimum, maximum, average length of different bouts/events explained

in this page. How could you measure the events with “> 2 sec” precision.

Also, how did you monitor and count/measure the no. of lordosis events in female mice; and the

no. of mount, intromission, and ejaculation events in male mice. If it’s manual monitoring, how

accurate is the counting.

Lines 170-172: In behavioral test, we have been using an event-recorder software (not commercial provided) to automatically measure the time and duration of events. The description of them was added in the Method. We did not calculate the average length in bouts (sorry), although raw data were stored in the software system. Calculating them requires some modification to the event-recorder programming.

e) lines 152-153: Give the IR formula in a proper mathematical formula/equation format. Refer

the comment 6b (page 7).

Lines 168-169: rewrote the equation.

9) page 10:

a) line 156: If the statement “IR was considered equally to LQ” was from the previously published

literature, cite the suitable reference for this. And if this is your own observation/statement, explain

why you considered this.

Lines 175-176: It is based on private communication with several researchers. Then, we modified the expressions of it. One example was cited.

b) line 157: Correct appropriately to “……..LQ, as clearly distinguished…….”, if needed.

Lines 175-176: corrected it.

c) lines 161-163: Why the female lordosis events, and so the true LQ, could not be

observed/measured accurately in the ordinary cage. How differently the bilevel chamber help the

experimenter to observe the female lordosis events accurately. Just to mention here, the male

mount, intromission, and ejaculation events are observed equally well in both the ordinary cage

and bilevel chamber apparatus.

And, if the true LQ cannot be measured in the ordinary cage test, why do you consider the

“estimated LQ”, measured based on the IR and %Intro, to compare with the true LQ and %Intro

calculated in the bilevel chamber test. Why don’t you use the “%Intro” in place of “estimated LQ”.

The sexual behavior of male mice has a characteristic movement pattern and is easily distinguished through visual observation. Conversely, assessing the presence or absence of female lordosis demands subtle judgment, considering factors such as tail position. In the ordinary cages, the direction of mouse bodies varied, making it challenging to consistently observe the angle of their tails.

d) lines 168-169: Correct the sentence appropriately to “……….and intromissions separately

revealed that females exhibited………….”.

Line 189: corrected it.

e) line 172: Replace “behavior” appropriately with “behavioral”, if needed.

Line 192: “behavior” was removed.

10) page 11:

a) lines 180-181: Why the female rejection-like behavior events could not be observed/measured

accurately in the ordinary cage. How differently the bilevel chamber help the experimenter to

observe the female rejection-like behavior events accurately.

Lines 121-125: We added the explanation why the female rejection-like behavior is hard to be observed in the ordinary cage in the Method.

b) line 182: Remove the word “in”.

Line 202: the sentence is modified.

11) page 12:

a) line 210: Use “and” before the word “dKO”.

Line 229: added “and”

b) line 210: Why is the “n” value difference between the WT, aKO, bKO, and dKO mice groups.

As demonstrating in this study, mice deficient in genes of AVP or OT receptors often faced challenges in reproduction, making it difficult to constantly obtain a specific number of mice. Consequently, we conducted behavioral tests simultaneously in subsets of subjects consisting of all experimental groups, and repeated the process. The variability in the number of mice among groups is a result of these circumstances.

12) page 13:

a) line 223: Does this line means that different stimulus males were used in the

subsequent/different bilevel chamber tests. Make it clear.

Lines 242-243: rewrote the sentence.

13) page 14:

a) lines 244-245: The LQ formula/equation on page 14, lines 244-245 is different from that on

page 7, lines 101-102. Correct this appropriately.

Lines 264-265: corrected it.

b) line 245: Replace “s” with “sec”, for seconds.

Lins 266: corrected it.

c) Correct the following:

line 245: Mean duration

Line 265: corrected it.

line 248: Total duration

Fig. 3B: Total duration

14) page 15:

a) lines 251-253: Explain it better. For example, in the test 1 also the lordosis duration was

suppressed in bKO females compared to WT females.

Lines 273-274: added a sentence to mentioned it.

b) line 259: Correct appropriately to “…post hoc analyses by Bonferroni test showed…….”.

Line 281: corrected it.

c) line 269: Figure title should be given in the Fig. 4 legend.

Lines 291-292: gave it a title.

15) page 16:

a) Correct the following:

line 270: Mean % time line 294: done.

line 275: Time line 298: done.

Fig. 4B: % time Fig. 4B: done.

b) Lines 280, 283: Why is the “n” value difference between the WT, OTRKO, aKO, bKO, and

dKO mice groups.

The same issue of Comment#11b

16) page 17:

a) line 296: Replace “s” with “sec”, for seconds.

Line 319: replaced it.

b) line 304: How do you explain the effect of test replication.

Lines 391-394: added the explanation about the effect of experience on rejection-like behavior.

17) page 18:

a) line 308: Figure title should be given in the Fig. 6 legend.

Lines 331-332: gave it a title.

b) Correct the following:

line 309: Mean % time line 334: done

line 314: Time line 338: done

Fig. 6B: % time Fig. 6B: done

c) What is the significance of % time spent with males, in the context of female sexual behavior.

As discussed in lines 398-402, during the apparatus design phase, we expected that % time spent sharing the same floor with males would be reflect of pacing behavior in female mice. However, upon carefully observation of OTRKO females during tests, which had an impact on this, we concluded that the elevated % time spent with males in OTRKO females resulted from prolonged interaction (rejection-like behavior) rather than changes in pacing behavior.

18) page 19:

a) line 328: If the v1b receptor is not essential for lordosis expression, then why bKO showed

almost no lordosis. Interpret this in a better way.

b) line 330: “in dKO females, aKO restored lordosis from the suppression by bKO up to the WT

level”. How do you justify this hypothesis. Any experimental evidence. And how the dKO showed

normal levels of lordosis, while the bKO showed no lordosis. It means there may be elements

(other than v1a and v1b receptors) involving in nullifying/inhibiting the effect of bKO in presence

of aKO. Can you shed light on this. If you propose that the crosstalk between OT and v1b receptors

(page 20, line 356) maybe the underlying cause for this, please explain it better. And if so, how

did you propose this hypothesis, and any attempts made for experimental evidence. Any other

elements may involve in this ?.

Lines 354-359: In response to your Comments 18a and 18b, we have further elaborated on our speculation in Discussion.

19) page 21:

a) How the OTRKO females exhibit normal lordosis levels, while showing increased rejection-like

behavior. What is the stimulus for lordosis in this case, if it’s not the male mount (which is

not possible due to rejection by female mice).

Lines 380-381: added a sentence about our explanation.

b) lines 378-379: Does the “rejection-like behavior” reflect “proceptivity”. Please explain this.

Lines 416-417: modified expression. The reduction of rejection-like behavior reflect “proceptivity”.

20) Have you attempted to evaluate the effects of AVP, OT, AVP/OT combination, progesterone

administrations/treatments in WT, aKO, bKO, dKO, OTRKO mice, similar to the rat lordosis

experiments (page 19, lines 333-334; page 20, lines 352-353; page 21, lines 371).

Lines 409-412: We added further prospects in study investigating the role of AVP and OT receptors on female sexual behavior.

21) Did you monitor and measure the Intromission and ejaculation events in experiments II and III.

Yes. We can add the data of male sexual behavior as the supplement.

22) Figure 2:

a) Replace “Ordinary” with “Ordinary Cage Test”, if appropriate

Replaced it.

b) Replace “Bilevel Chamber” with “Bilevel Chamber Test”, if appropriate

Replaced it.

23) P values and the corresponding bars should be given in the following figures.

a) Figure 3: Fig. 3B panel: P values and the corresponding bars for all the groups in Test 1.

P values appear in the Fig. legend.

b) Figure 4: Fig. 4B panel: P values and the corresponding bars for all the groups in Test 1 and 2.

P values appear in the Fig. legend.

c) Figure 5: Fig. 5A panel: P values and the corresponding bars for all the groups in Test 1 and 2.

P values appear in the Fig. legend.

To Reviewer 2:

Thank you very much for your evaluation. We appreciate your valuable advice immensely. Also, pointing out our mistakes has been incredibly helpful. We have revised the manuscript based on your comments, so please review it. The changes are highlighted, and each comment is numbered for your reference.

1) Figure 3A and b WT upper significance bar misleading, it looks like WT mice have statistical differences with both aKO and bKO, while the difference is only with bKO. Please check and modify the significance bars.

Thank you very much for pointing out our mistakes. These were corrected.

2) While the results of rejection-like behavior are clear, the influence of V1A and V1B receptors on lordosis seems puzzling. May you address more in the discussion why the absence of both receptors has the same results as their presence? While it seems to be some kind of interaction, it’s still unclear how knockout of V1AR can recover the effects of knockout of V1BR.

Lines 355-367: We included a description of our hypothesis deduced from the results.

3) Why you choose a different strain for experiment I? (C57BL6 for II and III is clear to me, because KO mice are based on the same strain)

Immediately after developing the bilevel chamber, we conducted a sexual behavior test using ICR female mice, known for their high sexual activity based on our experience, to verify its effectiveness. When preparing the paper, we recognized the importance of providing a thorough description of our new test equipment. Consequently, we decided to present the data as Experiment I, despite using a different mouse strain.

4) You used the terms receptivity and proceptivity in the conclusion and briefly used them in the discussion; May you clearly state in the introduction what you mean by those terms?

Lines 42-47: Thank you for your valuable advice. We delved more deeply into the concepts of "receptivity" and "proceptivity" in the Introduction.

5) May 

---

## [Decision Letter · Decision Letter 1]

1 May 2024

PONE-D-23-41361R1The bilevel chamber revealed differential involvement of vasopressin and oxytocin receptors in female mouse sexual behaviorPLOS ONE

Dear Dr. Kondo,

Thank you for submitting your manuscript to PLOS ONE. After careful consideration, we feel that it has merit but does not fully meet PLOS ONE’s publication criteria as it currently stands. Therefore, we invite you to submit a revised version of the manuscript that addresses the points raised during the review process.

The reviewer 1 provided a few minor comments that need to be addressed, apart from that, all looks good. Please submit your revised manuscript by Jun 15 2024 11:59PM. If you will need more time than this to complete your revisions, please reply to this message or contact the journal office at plosone@plos.org. Please include the following items when submitting your revised manuscript:A rebuttal letter that responds to each point raised by the academic editor and reviewer(s). You should upload this letter as a separate file labeled 'Response to Reviewers'.A marked-up copy of your manuscript that highlights changes made to the original version. You should upload this as a separate file labeled 'Revised Manuscript with Track Changes'.An unmarked version of your revised paper without tracked changes. You should upload this as a separate file labeled 'Manuscript'.If applicable, we recommend that you deposit your laboratory protocols in protocols.io to enhance the reproducibility of your results. Protocols.io assigns your protocol its own identifier (DOI) so that it can be cited independently in the future. For instructions see: https://journals.plos.org/plosone/s/submission-guidelines#loc-laboratory-protocols. Additionally, PLOS ONE offers an option for publishing peer-reviewed Lab Protocol articles, which describe protocols hosted on protocols.io. Read more information on sharing protocols at https://plos.org/protocols?utm_medium=editorial-email&utm_source=authorletters&utm_campaign=protocols.

We look forward to receiving your revised manuscript.

Kind regards,

Nirakar Sahoo, PhD

Academic Editor

PLOS ONE

Journal Requirements:

Reviewers' comments:

Reviewer's Responses to Questions

**Comments to the Author**

1. If the authors have adequately addressed your comments raised in a previous round of review and you feel that this manuscript is now acceptable for publication, you may indicate that here to bypass the “Comments to the Author” section, enter your conflict of interest statement in the “Confidential to Editor” section, and submit your "Accept" recommendation.

Reviewer #1: All comments have been addressed

Reviewer #2: All comments have been addressed

2. Is the manuscript technically sound, and do the data support the conclusions?

Reviewer #1: Yes

Reviewer #2: Yes

3. Has the statistical analysis been performed appropriately and rigorously? 

Reviewer #1: I Don't Know

Reviewer #2: Yes

4. Have the authors made all data underlying the findings in their manuscript fully available?

Reviewer #1: Yes

Reviewer #2: Yes

5. Is the manuscript presented in an intelligible fashion and written in standard English?

Reviewer #1: Yes

Reviewer #2: Yes

6. Review Comments to the Author

Reviewer #1: In the revised version of the manuscript entitled “The bilevel chamber revealed differential involvement of vasopressin and oxytocin receptors in female mouse sexual behavior”, the authors made several edits as per the reviewers’ comments. I appreciate that the authors addressed all the issues raised by the reviewers, made all the suggested/necessary corrections throughout the text and figures, which improved the manuscript a lot. I think that the newly included extended discussions at multiple places improved the general understandability of the manuscript. These insertions also shed light on the future prospective and directions of the research in mice sexual behavior, regulatory mechanism(s), and the AVP & OT roles. I hope the “Bilevel chamber” devised in this research will be a useful tool in the future research. And I believe in my understanding that the revised manuscript maybe in good shape for publication with the following few minor modifications.

Minor Comments:

Line 56: Reframe the sentence to correct “with of nine”.

Line 95: Remove the extra space before the word “two”.

General comment: Please correct the page numbers in the responses to the reviewers’ comments, if the author responses along with the reviewer comments will be published as a separate section in the published manuscript.

Reviewer #2: The authors did a good job of addressing my comments and giving satisfactory answers. Overall I recommend accepting the manuscript at this stage.

7. PLOS authors have the option to publish the peer review history of their article (what does this mean?). If published, this will include your full peer review and any attached files.

Reviewer #1: **Yes: **MADHUSUDHANARAO KATIKI

Reviewer #2: No

---

## [Author Response · Author response to Decision Letter 1]

4 May 2024

To Reviewer 1:

Thank you sincerely for your kind and detailed review. We made the correction following your comments.

We believe that our corrections meet your criterion.

Reviewer’s comments

Line 56: Reframe the sentence to correct “with of nine”.

Thank you. We rewrote the sentence.

Line 95: Remove the extra space before the word “two”.

Thanks again. We removed the extra one.

---

## [Editor Report · Decision Letter 2]

17 May 2024

The bilevel chamber revealed differential involvement of vasopressin and oxytocin receptors in female mouse sexual behavior

PONE-D-23-41361R2

Dear Dr. Kondo,

We’re pleased to inform you that your manuscript has been judged scientifically suitable for publication and will be formally accepted for publication once it meets all outstanding technical requirements.

Kind regards,

Nirakar Sahoo, PhD

Academic Editor

PLOS ONE
---

## [Editor Report · Acceptance letter]

27 May 2024

PONE-D-23-41361R2 

PLOS ONE

Dear Dr. Kondo, 

I'm pleased to inform you that your manuscript has been deemed suitable for publication in PLOS ONE. Congratulations! Your manuscript is now being handed over to our production team.

Kind regards, 

on behalf of

Dr. Nirakar Sahoo 

Academic Editor

PLOS ONE